# In Vitro Anti-Inflammatory Activity and Structural Characteristics of Polysaccharides Extracted from *Lobonema smithii* Jellyfish

**DOI:** 10.3390/md21110559

**Published:** 2023-10-26

**Authors:** Thitikan Summat, Sutee Wangtueai, SangGuan You, Weerawan Rod-in, Woo Jung Park, Supatra Karnjanapratum, Phisit Seesuriyachan, Utoomporn Surayot

**Affiliations:** 1College of Maritime Studies and Management, Chiang Mai University, Samut Sakhon 74000, Thailand; thitikan.ts@gmail.com (T.S.); sutee.w@cmu.ac.th (S.W.); 2Department of Marine Bio Food Science, Gangneung-Wonju National University, Gangneung 25457, Gangwon, Republic of Korea; umyousg@gwnu.ac.kr (S.Y.); weerawan.ve@gmail.com (W.R.-i.); pwj0505@gwnu.ac.kr (W.J.P.); 3East Coast Life Sciences Institute, Gangneung-Wonju National University, Gangneung 25457, Gangwon, Republic of Korea; 4Department of Agricultural Science, Faculty of Agriculture Natural Resources and Environment, Naresuan University, Phitsanulok 65000, Thailand; 5Division of Marine Product Technology, Faculty of Agro-Industry, Chiang Mai University, Chiang Mai 50100, Thailand; supatra.ka@cmu.ac.th; 6Faculty of Agro-Industry, Chiang Mai University, Chiang Mai 50100, Thailand; phisit.s@cmu.ac.th

**Keywords:** *Lobonema smithii*, jellyfish, polysaccharides, macrophages, anti-inflammatory

## Abstract

Crude polysaccharides were extracted from the white jellyfish (*Lobonema smithii*) using water extraction and fractionated using ion-exchange chromatography to obtain three different fractions (JF1, JF2, and JF3). The chemical characteristics of four polysaccharides were investigated, along with their anti-inflammatory effect in LPS-stimulated RAW264.7 cells. All samples mainly consisted of neutral sugars with minor contents of proteins and sulphates in various proportions. Glucose, galactose, and mannose were the main constituents of the monosaccharides. The molecular weights of the crude polysaccharides and the JF1, JF2, and JF3 fractions were 865.0, 477.6, 524.1, and 293.0 kDa, respectively. All polysaccharides were able to decrease NO production, especially JF3, which showed inhibitory activity. JF3 effectively suppressed iNOS, COX-2, IL-1β, IL-6, and TNF-α expression, while IL-10 expression was induced. JF3 could inhibit phosphorylated ERK, JNK, p38, and NF-κB p65. Furthermore, flow cytometry showed the impact of JF3 on inhibiting CD11b and CD40 expression. These results suggest that JF3 could inhibit NF-κB and MAPK-related inflammatory pathways. The structural characterisation revealed that (1→3)-linked glucopyranosyl, (1→3,6)-linked galactopyranosyl, and (1→3,6)-linked glucopyranosyl residues comprised the main backbone of JF3. Therefore, *L. smithii* polysaccharides exhibit good anti-inflammatory activity and could thus be applied as an alternative therapeutic agent against inflammation.

## 1. Introduction

The polysaccharides of marine organisms, which are bioactive natural products with medicinal properties, have been shown to possess a large range of biological properties, such as antiviral, antioxidant, antitumoral, anti-cancer, anti-obesity, and cardioprotective activities [1,2,3,4,5], especially in immune responses [4,5,6,7]. Marine-derived polysaccharides can also be extensively used in the field of biomedical engineering [8]. Recent studies have reported on the anti-inflammatory activities of natural polysaccharides against stimulated inflammation in RAW264.7 macrophages [9,10,11,12]. Lipopolysaccharides (LPS) are a major component of the cell wall of Gram-negative bacteria and are one of the most potent inflammatory agents [10]. Indeed, LPS-stimulated macrophages are a useful model for studying inflammation, possible anti-inflammatory agents, and their action mechanisms [13,14]. Numerous factors can be used for monitoring the anti-inflammatory effects of natural products in in vitro cell culture models, such as inflammatory cytokines (tumour necrosis factor-α (TNF-α), interleukin-1β (IL-1β), IL-4, IL-6, and IL-10), inflammatory enzymes such as nitric oxide (NO), nitric oxide synthases (iNOS), prostaglandin E2 (PGE_2_), cyclooxygenase-2 (COX-2), and cell surface receptor proteins (CD11b and CD40), which contribute to modulating the inflammatory response and process [10,14,15,16].

Jellyfish are marine invertebrate animals consisting of around 200 described species of the class Scyphozoa of the phylum Cnidaria [17]. There are several species of edible jellyfish, including *Lobonema smithii*, *Rhopilema hispidum*, *Rhopilema esculentum*, *Nemopilema nomurai*, and *Lobonemoides gracilis* [18]. Jellyfish contain a variety of nutrients, including protein, amino acids, carbohydrates, vitamins, and inorganic elements, making them valuable economic and nutritional resources [7,19]. Previous studies have shown that jellyfish polysaccharides isolated from *R. esculentum* have anti-inflammatory, antioxidant, and immunomodulatory activities in macrophages and C57BL/6 mice [7,19,20]. An aqueous extract of the jellyfish *N. nomurai* exhibited an anti-inflammatory effect on RAW264.7 macrophages activated by LPS and a zebrafish model [21].

The white jellyfish (*L. smithii*) belongs to the family Lobonematidae, which is an important fishery commodity in Southeast Asia [22]. This jellyfish mainly consists of collagen protein [23,24]. The protein hydrolysate of *L. smithii* has been revealed to possess antioxidant, antibacterial, and tyrosinase inhibitory activity [25,26]. However, information on the anti-inflammatory properties of polysaccharides from *L. smithii* has not yet been reported. In this study, we isolated and fractionated polysaccharides from *L. smithii* before investigating the protective action of *L. smithii* on inflammation triggered by LPS and probable signalling pathways implicated through in vitro studies using RAW264.7 cells. Furthermore, the polysaccharides were also evaluated physiochemically and structurally.

## 2. Results

### 2.1. Chemical Composition of Polysaccharides Isolated from L. smithii

The yields of polysaccharides from *L*. *smithii* as crude polysaccharides and as fractions are presented in Table 1. The yield of crude polysaccharides was 1.3%, which mainly consisted of carbohydrates (62.5%) and proteins (24.1%) with minor amounts of sulphates (10.2%) and uronic acids (3.17%). Analysis of monosaccharide composition showed that L-arabinose (15.0%), D-mannose (15.2%), D-glucose (33.1%), and D-galactose (26.2%) were the predominant sugars in the crude polysaccharides, with significant levels of L-rhamnose (8.30%) and L-fucose (2.23%), indicating a heterogeneous monosaccharide composition.

The crude polysaccharides of *L. smithii* were further separated on a DEAE Sepharose column, producing three different fractions, JF1, JF2, and JF3. The elution yields of JF1 (distilled water), JF2 (1.0 M NaCl), and JF3 (1.5 M NaCl) were 77.2%, 13.2%, and 9.7%, respectively. All fractions contained carbohydrates as the major component, while the protein, sulphate, and uronic acid contents varied. The crude polysaccharides and all fractions contained carbohydrates as the major component (62.5%), while protein (24.1%) was present in considerable amounts in the crude polysaccharides. Fraction JF1 possessed a chemical composition comparable to the crude polysaccharides (carbohydrate 72.2% and protein 18.1%), while the JF2 fraction contained carbohydrates (66.0%) alongside proteins (13.6%), sulphates (17.3%), and lesser amounts of uronic acid (2.97%). On the other hand, carbohydrates (67.6%), as well as considerable amounts of sulphates (22.7%) and proteins (7.12%), were included in the JF3 fraction. The uronic acid contents of all fractions were minor. The monosaccharide composition of the JF1 fraction consisted of D-galactose (41.0%) and D-glucose (40.2%), which were the main sugars in the fraction, alongside considerable amounts of L-arabinose (11.7%). The majority of the unit sugars in the JF2 fraction also comprised D-glucose (52.4%) and minor amounts of L-arabinose (15.4%), D-galactose (12.5%), and D-mannose (10.6%), as well as minor levels of L-rhamnose (6.36%) and L-fucose (2.53%). The JF3 polysaccharides contained D-glucose, D-galactose, and D-mannose (56.7%, 28.4%, and 13.7%, respectively) as the main sugars, while L-arabinose, L-rhamnose, and L-fucose represented minor sugar units (0.60%, 0.54%, and 0.10%, respectively). The separation of crude polysaccharides using a DEAE Sepharose fast flow column indicated that the different fractions of polysaccharides were characterised by different ionic strengths and chemical properties.

### 2.2. Molecular Weight (M_w_) Analysis

Figure 1 displays the combined UV and RI chromatograms of the crude and fractions. A significant amount of crude polysaccharide was eluted from the high-performance size-exclusion chromatography (HPSEC) column between elution times, with broad primary peaks revealing their heterogeneous polymer distributions (Figure 1A). According to the UV chromatogram, proteins were found in peak quantity at an elution duration of 25 to 40 min. The M_w_ of the peaks obtained with the MALLS technique was 865.0 kDa (Table 2). The estimated radius of gyration (R_g_) from the peaks also served as a method of approximating the size of the crude polysaccharides (Table 2). The R_g_ values of the peaks were 124.0 nm. Crude polysaccharides showed a significant UV chromatogram, in agreement with the chemical constitution of their 24.1% protein content (Table 1).

The chromatograms of the fraction profiles showed a single dominant peak for JF1 at the elution duration ranging from 27.3 to 45.2 min, as well as a major peak at 25.6 to 45.1 min for JF2 (Figure 1B,C, respectively). However, the elution time for JF3 ranged between 28.5 and 40.0 min, exhibiting a narrower peak than the other fractions, thus indicating that the polymer was more homogeneous (Figure 1D).

The relatively low value of the UV chromatogram compared with the marked RI signal in fractions JF2 and JF3 suggested the presence of small quantities of protein in these fractions. Conversely, the significant UV reaction shown in the JF1 fraction was consistent with its 18.1% protein level. The M_w_ values of JF1, JF2, and JF3 were significantly less than those of the crude polysaccharides. Table 2 shows that the M_w_ values were 477.6, 524.1, and 293.0 kDa for JF1, JF2, and JF3, respectively. It was previously reported that the M_w_ values of jellyfish skin polysaccharides (RP-JSP1 and RP-JSP2) were 121 and 590 kDa, respectively [27]. On the other hand, the M_w_ values differed from those reported by Li et al., who found that the M_w_ of jellyfish polysaccharides (JSP1) was 1250 kDa [7]. R_g_ is explained as the distribution of units of a polysaccharide around its axis and can define the size of the polymer, which can be calculated from light scattering up to various angles of the MALLS system. Table 2 shows that the R_g_ values were 124.0, 77.7, and 94.4 nm for the crude, JF1, and JF2 fractions and 56.3 nm for fraction JF3, indicating that in this result, the size of the polymer was in accordance with the M_w_.

The individual volume for gyration (SV_g_) of the SPs may be derived from the values of M_w_ and R_g_ using the following equation, as described by You and Lim [28].
$${\mathrm{SV}}_{g}=\;\frac{\frac{4}{3}\pi \;{\left(R_{g}\times 10^{8}\right)}^{3}}{M_{w}\;/\;N}\;=\;\frac{2.522{\;R}_{g}^{3}}{M_{w}}$$
where the values for SV_g_, M_w_, and R_g_ are cm^3^/g, g/mol, and nm, respectively, and N is Avogadro’s number (6.02 × 10^23^/mol). SV_g_ is inversely related to the degree of molecular compactness, giving the theoretical gyration volume per unit of molar mass and providing mass-based information on polysaccharide density [28]. The overall SV_g_ values of the crude and fractionated polysaccharides are shown in Table 2. The SV_g_ values for the crude polysaccharides were 5.57 cm^3^/g, and the values for their fractions were 5.57, 2.49, 4.05, and 1.59, respectively. The crude SV_g_ values were significantly higher than those of the fractions. This finding indicated that the crude possessed less compact and more extended conformational structures than JF2, following JF1 and JF3, which were the most compact.

These discrepancies in M_w_, R_g_, and SV_g_ were most likely caused by a combination of factors, including the extraction and purification processes as well as the analytical methods used in each investigation. 

### 2.3. Effects of L. smithii Polysaccharides on the Cell Viability and NO Production

This study evaluated the anti-inflammatory effects of crude and fractionated (JF1, JF2, and JF3) polysaccharides isolated from *L. smithii*. Initially, the potential cytotoxicity of *L. smithii* polysaccharides on RAW264.7 cells, varying from 125 to 1000 μg/mL, was examined using a WST assay. As shown in Figure 2A, *L. smithii* polysaccharides significantly increased cell proliferation up to 1000 μg/mL relative to the negative control (RPMI medium). This finding suggests that the polysaccharides were not cytotoxic to RAW264.7 cells at the concentrations tested.

In addition, the immunoinhibiting effect was preliminarily evaluated by measuring NO accumulation (Figure 2B). All *L. smithii* polysaccharides decreased NO production in a dose-dependent manner. Among the polysaccharides, fraction JF3 showed a higher production of NO than the other polysaccharides. Increasing the concentrations of JF3 (125 to 1000 μg/mL) enhanced NO generation by 10.05 ± 0.45 μM, while the same concentrations of the crude polysaccharides, JF1, and JF2 reduced the NO production by 33.94 ± 0.54, 32.45 ± 0.45, and 31.82 ± 1.08 μM, respectively. According to these results, the most immunoinhibitory fraction, JF3, was chosen from the four different *L. smithii* polysaccharides to be assessed for their anti-inflammatory activity on macrophages at concentrations ranging from 125 to 1000 μg/mL.

### 2.4. JF3 Inhibits LPS-Induced Cell Viability and NO Production

To further investigate the effect of the most immunoinhibiting JF3 on the inhibition of macrophage activation, RAW264.7 cells were pretreated with or without JF3 (125–1000 μg/mL) and aspirin (200 µg/mL, as a positive drug) before being induced with LPS (1 μg/mL). As shown in Figure 3A, fraction JF3 (125–1000 μg/mL) significantly enhanced macrophage viability in RAW264.7 cells, and the enhancement increased with the JF3 concentration. Importantly, JF3 did not cause any significant toxicity to these cells. Therefore, these doses were used in the following experiments to treat JF3. As shown in Figure 3B, the NO secretion levels in the RAW264.7 cells were measured to be 0.55 ± 0.05 μM in the negative control and 37.70 ± 0.67 μM in the LPS-induced cells alone. In the presence of JF3 at concentrations of 125 to 1000 μg/mL, the levels of NO production were significantly inhibited, reducing the NO secretion levels to 35.31 ± 1.34, 31.94 ± 0.27, 23.28 ± 0.83, and 9.66 ± 0.72 μM. In addition, the NO contents at 1000 μg/mL JF3 concentrations were lower than those of the positive drug (aspirin).

### 2.5. JF3 Inhibits LPS-Induced Expression of iNOS, COX-2, and Cytokines

To determine whether the JF3 fraction inhibited the LPS-induced inflammatory response, the levels of inflammatory mediators (iNOS and COX-2) and inflammatory cytokines (IL-1β, IL-6, IL-10, and TNF-α) were measured. As shown in Figure 4A,B, the mRNA expression levels of iNOS and COX-2 were reduced in a dose-dependent manner by JF3 in the LPS-induced RAW264.7 cells. In addition to inhibiting the iNOS and COX-2 expressions, a dose-dependent suppression of JF3 on the IL-1β, IL-6, and TNF-α expression was observed (Figure 4C–E). Meanwhile, the effects of JF3 on IL-10 expression in the LPS-induced RAW264.7 cells were evaluated. As shown in Figure 4F, the mRNA level of IL-10 expression was dramatically raised following JF3 treatment at dosages ranging from 125 to 1000 μg/mL. These results show that the JF3 fraction activated the expression of inflammatory mediators and cytokines at the transcriptional level.

### 2.6. JF3 Suppresses LPS-Induced Nuclear Factor-κB (NF-κB) Activation

To investigate the possible inhibitory effects of JF3 through the suppression of LPS-induced activation of NF-κB signalling, the effect of JF3 on the nuclear translocation of the NF-κB p65 subunit was assessed using Western blot analyses. Figure 5A shows that LPS stimulation induced the phosphorylation of the NF-κB-p65 subunit. The JF3 fraction dramatically reduced phosphorylated NF-κB-p65, depending on the concentration. These findings imply that JF3 may prevent LPS-induced NF-κB activation.

### 2.7. JF3 Suppresses LPS-Induced Mitogen-Activated Protein Kinase (MAPK) Activation

It is well known that MAPK signalling pathways play an essential role in LPS-activated RAW264.7 cells [14,29,30]. The effects of JF3 on the phosphorylation levels of extracellular signal-regulated kinase (ERK), c-Jun N-terminal kinase (JNK), and p38 MAPK were determined in LPS-stimulated cells. As shown in Figure 5B, all MAPKs were stimulated by LPS treatment, whereas JF3 suppressed the LPS-induced phosphorylation of ERK, JNK, and p38 MAPK. Moreover, JF3 (125–1000 μg/mL) was shown to suppress the phosphorylation of three MAPKs in a dose-dependent manner, suggesting that MAPK pathways might contribute to the anti-inflammatory effects of JF3.

### 2.8. JF3 Inhibits LPS-Induced Cell Surface Expression

Fluorescence-activated cell sorting (FACS) analysis was utilised to investigate the effects of JF3 on LPS binding to the surface of RAW264.7 cells. As shown in Figure 6A,B, JF3 exerted an inhibitory effect on cell surface molecules such as CD11b and CD40. LPS treatment increased both cell surface molecules compared with RPMI. Compared with LPS, the LPS-induced CD11b expression was significantly reduced by JF3 at 125–1000 μg/mL in a dose-dependent manner, whereas the LPS-induced CD40 expression showed no differences in significance (*p* < 0.05) at 125 μg/mL concentrations and was substantially decreased at doses ranging from 250 to 1000 µg/mL.

### 2.9. Methylation Analysis of JF3

The glycosidic linkage of the most anti-inflammatory JF3 polysaccharides was analysed by GC–MS. Table 3 reveals that the alditol acetate products of JF3 mainly consisted of 2,4,6-Me3-Glc*p*, 2,4-Me2-Glcl*p*, and 2,4-Me2-Gal*p,* indicating the presence of (1→3)-linked glucopyranosyl, (1→3,6)-linked glucopyranosyl, and (1→3,6)-linked galactopyranosyl residues. A small amount of 2,3,4-Me3-Gal*p* was also detected, suggesting a (1→6)-linked galactopyranosyl residue. In addition, JF3 also included 2,3,4,6-Me4-Man*p*,** implying a terminal residue. After desulphating fraction JF3 (D-JF3) by the solvolytic desulphation method, the methylation analysis of D-JF3 exhibited a marked decrease in 2,4-Me2-glucitol acetate with a concomitant increase in 2,4,6-Me3-glucitol acetate and no changes in the proportions of other methylated alditol acetates (Table 3). This suggests that the sulphate groups were mostly linked at the O-6 position. Overall, the above data indicate that the polysaccharide chain JF3 could be a (1→3)-glucose partially sulphated at O-6. Furthermore, (1→6)-linked galactose could be a branched polysaccharide, and other units and terminals could be in the side chains. Future studies need to elucidate the connection between the linkage through 1D and 2D NMR analyses. Li et al. (2017) reported the linear linkage of the fractionated polysaccharides of jellyfish with a M_w_ of 1250 kDa (JSP-11), which was previously reported as (1→3,6)-Man*p*, (1→6)-Gal*p*, and (1→)-Glc*p*A as terminal [7].

## 3. Discussion

This study focused on the extraction, isolation, structure characterisation, and anti-inflammation properties of polysaccharides from the white jellyfish (*L. smithii*). After fractionation using ion-exchange chromatography to obtain three fractions (JF1, JF2, and JF3), the crude and three fractions were found to contain carbohydrates (62.5–72.2%) with some proteins (7.12–24.1%), sulphate (8.27–22.7%), and uronic acids (1.47–3.40%). The M_w_ and R_g_ of these polysaccharides showed ranges of 293.0–865.0 kDa and 56.3–124.0 nm, respectively, indicating variations in their molecular size. Moreover, the monosaccharide composition revealed differences in levels, including galactose (12.5–41.0%), glucose (33.1–56.7%), mannose (3.83–15.2%), and arabinose (0.60–15.4%), with a small amount of fucose (0.10–2.53%) and rhamnose (0.54–8.30%). The backbones of the most immunoinhibiting polysaccharide, JF3, were mainly linked through (1→3)-glucose, with some sulphate groups attached at position O-6. In addition, there is a possibility that (1→6)-linked galactose could form a branching structure within a polysaccharide, and various units and terminals could be present in the side chains.

The research of hot water extraction of polysaccharides from *Gracilaria rubra* (GRPS) and isolation with a DEAE-52 cellulose column and a Sephadex G-50 column yielded three fractions, GRPS-1-1, GRPS-2-1, and GRPS-3-2, which were heteropolysaccharides consisting of galactose and fucose at ratios of 1.79:1, 2.16:1, and 2.76:1, respectively. In addition, the M_w_ of GRPS-1-1, GRPS-2-1, and GRPS-3-2 were 1310, 691, and 923 kDa, respectively [31]. Another study performed enzyme-assisted extraction of sulphated polysaccharide (SCVP-1) from the sea cucumber, which consisted of mannose, glucosamine, glucuronic acid, *N*-acetyl-galactosamine, glucose, galactose, and fucose; had a relative molecular weight of 180.8 kDa; and was composed of total carbohydrates, uronic acid, proteins, and sulphate groups. The structure showed glycosaminoglycan with sulphation linked to the fucose residue [32]. In general, the structure and molecular properties of polysaccharides differ greatly depending on the type, species, growing conditions, extraction process, and analytical techniques. Among these parameters, the variation in their proximate compositions appears to be significantly impacted by the type, species variations, and extraction processes [33,34].

Numerous polysaccharides have been demonstrated to possess anti-inflammatory effects in macrophages, suppressing the production of inflammation mediators, such as NO, PGE_2_, iNOS, COX-2, IL-1β, IL-6, and TNF-α [9,11,31,35]. The inflammation process involves multiple molecular mechanisms. For example, iNOS and COX-2 are two of the most important mediators of NO production and prostaglandin synthesis [11,13]. First, we observed that *L. smithii* polysaccharides, both in crude form and when fractionated (JF1, JF2, and JF3), could effectively inhibit the LPS-induced production of NO; the highest inhibitory effect on NO production was shown by fraction JF3. Our results demonstrate that JF3 inhibited LPS-induced NO production in macrophages, which was associated with their ability to downregulate iNOS mRNA expression. TNF-α, IL-1β, and IL-6 are pro-inflammatory cytokines that play essential roles in immunological responses to a range of inflammatory stimuli [36]. JF3 also significantly reduced the expression of COX-2, IL-1β, IL-6, and TNF-α induced by LPS. Conversely, IL-10 is one of the most effective anti-inflammatory cytokines, with numerous immunomodulatory properties [10,12,37]. Our findings revealed that JF3 dramatically boosted the levels of IL-10, an anti-inflammatory cytokine. Furthermore, these results are consistent with those of previous reports, which showed that polysaccharides regulated LPS-induced inflammatory mediators and cytokine production [38,39,40]. Overall, this study suggests that JF3 polysaccharides have anti-inflammatory activity by inhibiting the expression of pro-inflammatory cytokines and enhancing anti-inflammatory cytokines.

A variety of signalling pathways related to cellular immunological responses have been studied to elucidate the molecular mechanisms of *L. smithii* polysaccharides. Polysaccharides may be involved in immunity and inflammation through MAPK and NF-κB signalling processes, which are critical intracellular signalling pathways with complicated connections [15]. NF-κB is an important transcription factor that regulates the generation of various inflammation-related mediators and cytokines [36,39]. Here, pretreatment with JF3 significantly reduced LPS-induced p65 phosphorylation. The MAPK pathway is additionally recognised to serve as a critical mediator in the promotion of inflammatory responses, which plays a role in regulating several cellular processes [41]. The pathway consists of three-tiered cascades, including JNK, ERK, and p38 [10,42]. The phosphorylation activates pro-inflammatory transcription factors, such as activator proteins (AP)-1 (cFos/cJun), Runt-related transcription factor (RUNX)-2, hypoxia inducible factor (HIF)-2α, and CCAAT-enhancer-binding protein (C/EBP)-β [42]. In this work, JF3 markedly decreased the phosphorylation of p38, JNK, and ERK1/2. A previous study found that sulphated polysaccharides from *Sargassum cristaefolium* had anti-inflammatory effects by suppressing NF-κB and downregulating the phosphorylation of p-38, ERK, and JNK kinases in the MAPK signalling cascade [6].

Additionally, markers such as CD68, CD86, CD80, F4/80, CD11b, and CD40 are expressed by activated macrophages [43,44]. The CD40 molecule interacts with antigen-presenting cells and plays a costimulatory role in immune regulation [16]. CD11b/CD18 is a receptor on innate immune cells that recognises pathogens [45]. Here, the flow cytometric analysis revealed that JF3 inhibited the expression of CD11b and CD40 in LPS-treated RAW264.7 cells. Thus, our findings suggest that JF3 could inhibit the initial phase of LPS-activated cellular signalling pathways by decreasing LPS binding to cell surface receptors and CD11b and CD40 expression on RAW264.7 cells.

Recently, research demonstrated that the structural characteristics of a polysaccharide have a close relationship with anti-inflammatory activities, including chemical composition, monosaccharide, molecular weight, structure, functional groups, glycosidic linkage, and conformations. Many reports have shown that the sulphate groups and M_w_ of polysaccharides markedly affect their anti-inflammatory properties [46]. Brown algal polysaccharides (*S. cristaefolium*) with an M_w_ of 386.1 kDa and a sulphate concentration of 9.42% reduced NO production in LPS-stimulated RAW264.7 cells [6]. It seems that the dependence of suitable M_w_ and sulphate groups might enhance the binding affinity with the cell receptor [6,47].

According to our results, the JF3 chain contained (1→3)-linked glucopyranosyl, (1→3,6)-linked glucopyranosyl, and (1→3,6)-linked galactopyranosyl residues. The sulphate groups were mostly presented at the position of O-6. In addition, the 1→6-linked galactose units could be a branched polysaccharide, and other units and terminals could be in the side chains. JF3 had a molecular weight of 293.0 kDa and a high content of sulphates. This is similar to the previously mentioned research, which found that glycosidic linkages could affect anti-inflammatory activities. α-D-(1→3)-linked glucose shows anti-inflammatory, antiangiogenic activities, and anti-CAG (chronic atrophic gastritis) activity [48,49,50,51]. Similarly, the existence of monosaccharides could be a factor that affects the anti-inflammatory activities of polysaccharides, glucose, and fucose, and has been found to have a good effect on inflammatory activity [5,52]. Numerous studies have been conducted on a fucose-rich fucoidan that strongly affects both in vitro and in vivo anti-inflammatory activities. In brown seaweed, the fucoidan’s effects could potentially inhibit anti-inflammatory activities via the inhibition of protein denaturation, primarily due to its fucose content and the moderation of sulphate content [53,54]. An analysis of monosaccharides of JF3 revealed that they mostly contained glucose, galactose, and mannose, while arabinose, rhamnose, and fucose represented a minor sugar unit. However, the limited quantities of fucose may be responsible for the anti-inflammatory activities. As such, investigation into the monosaccharide, molecular weight, and fine structures is essential to understand the relationships between structural elements and biological processes and realise their maximum effects.

## 4. Materials and Methods

### 4.1. Materials and Chemicals

Fresh white jellyfish (*L. smithii*) were obtained from local fisheries in La-ngu District, Satun Province, Thailand. After collection, fresh jellyfish were individually packed into a polyethylene bag, frozen in the freezer, packed into a carton box, and transported by a temperature-controlled container car (−18 to 20 °C) to the laboratory of the College of Maritime Studies and Management, Chiang Mai University, Samut Sakhon Province, Thailand, within 12 h. The collected jellyfish was rinsed with distilled water (DW) and dried in a hot air oven at 55 °C until the final moisture content of the sample was 8%. The dried sample was then powdered with a grinder. The jellyfish was placed into a polyethylene bag, frozen at −20 °C, and used for polysaccharide extraction after two months. The Roswell Park Memorial Institute (RPMI) 1640 Medium was purchased from Gibco (Thermo Fisher Scientific Inc., Waltham, MA, USA). Mouse macrophage RAW264.7 cells were purchased from the Korean Cell Line Bank (KCLB; Seoul, Republic of Korea). Fetal bovine serum (FBS) and 1% penicillin/streptomycin were procured by Welgene (Daegu, Republic of Korea). All other cell culture chemicals and reagents were obtained from Sigma Chemical Co. (St. Louis, MO, USA). All solvents used were of high-performance liquid chromatography (HPLC) grade.

### 4.2. Polysaccharide Extraction and Fractionation

The extraction of *L. smithii* polysaccharides was performed using water extraction [7]. The dried sample powder of *L. smithii* was extracted two times in DW at 90 °C for 2 h. All water extracts were combined and precipitated from the supernatant using four volumes of cold 98% (*v*/*v*) ethanol and stored overnight at 4 °C, which was then subjected to filtration. The resulting pellet was collected and dried under a fume hood overnight to obtain the crude polysaccharide powder. The crude polysaccharide was then fractionated using ion-exchange chromatography on a DEAE Sepharose fast flow column (17-0709-01; GE Healthcare Bio-Science AB, Uppsala, Sweden). The polysaccharides were eluted with DW and different concentrations of NaCl (0.5–2 M). The elution profile of carbohydrates was evaluated using the phenol-H_2_SO_4_ method by determining 490 nm absorbance [55]. The chromatography yielded three fractions, namely, JF1 (eluted with DW), JF2 (eluted with 1.0 M NaCl), and JF3 (eluted with 1.5 M NaCl), as shown in Figure 7.

### 4.3. Chemical Composition

The chemical compositions were examined, including the total carbohydrate content, which was determined with the phenol-sulfuric acid method using D-glucose as a standard [55]. The protein content was evaluated with the Lowry method using a DC protein assay kit (Bio-Rad, Hercules, CA, USA) [56]. The determination of the sulphate content was carried out with the BaCl_2_ gelatin method using K_2_SO_4_ as a standard [57]. The uronic acid contents were analysed with a sulphamate/m-hydroxydiphenyl assay using glucuronic acid as a standard [58].

### 4.4. Monosaccharide Characterisation

Gas chromatography–mass spectrometry (GC–MS) was used to examine the monosaccharide composition of *L. smithii* polysaccharides. The sample was hydrolysed with trifluoroacetic acid (TFA, 4 M) at 100 °C for 6 h, then reduced in water with sodium borodeuteride (NaBD_4_) and acetylated with acetic anhydride. Finally, the sample was analysed by GC–MS (6890 N/MSD 5973, Agilent Technologies, Santa Clara, CA, USA) coupled with an HP-5MS capillary column (30 m × 0.25 mm × 0.25 µm). As a carrier gas, nitrogen was applied. To certify the monosaccharide content, monosaccharide standards were employed.

### 4.5. Measurement of M_w_

The M_w_ of the four polysaccharides was estimated using the HPSEC-UVMALLS-RI system), which included a pump (Waters 510, Milford, MA, USA), an injector valve with a 200 μL sample loop (model 7072, Rheodyne, Rohnert Park, CA, USA), SEC columns (TSK G5000 PW, 7.5 mm × 600 mm; TosoBiosep, Mongomeryville, PA, USA), a UV detector at 280 nm (Waters 2487), a multi-angle laser light scattering detector (HELEOS, Wyatt Technology Corp, Santa Barbara, CA, USA), and a refractive index detector (Waters 2414). At a flow rate of 0.4 mL/min, the mobile phase of this system contained 0.15 M NaNO_3_ and 0.02% NaN_3_. Each sample was immersed in DW and then heated for 15 min at 75 °C before being injected into MALLS. The M_w_ and R_g_ were estimated using the ASTRA version 6.0 software (Wyatt Technology Corp., Beijing, China).

### 4.6. Measurement of the Anti-Inflammatory Activity of L. smithii Polysaccharides

#### 4.6.1. Cell Culture and Treatment

The RAW264.7 cells were grown in RPMI-1640 medium, which was supplied with 10% FBS and 1% penicillin/streptomycin and kept at 37 °C in 5% CO_2_ humidified incubators. The cells were incubated with 100 µL of various doses of polysaccharides (125, 250, 500, and 1000 µg/mL) or aspirin (200 µg/mL) as a positive drug for 1 h. Following sample treatment, the cells were stimulated with 100 µL of LPS (1 µg/mL) for a further 24 h.

#### 4.6.2. Cell Viability Analysis

The cell proliferative ability was assessed through the EZ-Cytox Cell Viability Assay Kit (DaeilLab Service, Seoul, Republic of Korea). Briefly, the RAW264.7 cells (1 × 10^6^ cells/mL) were incubated in a 96-well cell culture plate with various concentrations of *L. smithii* polysaccharides for 24 h, and the amount of tetrazolium salt was quantified using a microplate reader (EL-800; BioTek Instruments, Winooski, VT, USA) at 450 nm absorbance.

#### 4.6.3. Determination of NO Release

Griess reagent (Sigma-Aldrich, St. Louis, MO, USA) was applied to quantify the concentration of NO in the supernatant. RAW264.7 cells (1 × 10^6^ cells/mL) in a 96-well cell culture plate were subjected to treatment with *L. smithii* polysaccharides prior to LPS stimulation. After incubation, 100 μL of cell culture supernatant was combined with Griess reagent (0.1% *N*-1-napthyl ethylenediamine dihydrochloride in DW and 1% sulphanilamide in 5% phosphoric acid) and incubated for 10 min. The nitrite accumulation was measured using 540 nm absorbance and a standard curve for sodium nitrite.

#### 4.6.4. Real-Time Polymerase Chain Reaction (PCR) Analysis

The RAW264.7 cells (1 × 10^6^ cells/mL) were placed in a 24-well cell culture plate and subjected to treatment with polysaccharides induced by LPS. Total RNA extraction was carried out using the TRIzol reagent (Invitrogen, Carlsbad, CA, USA), and cDNA was created using a High-Capacity cDNA Reverse Transcription Kit (Applied Biosystems, Foster City, CA, USA). Then, the fragments were amplified using a real-time PCR system on a QuantStudio™ 3 FlexReal-Time PCR System (Thermo Fisher Scientific Inc., Waltham, MA, USA) with TB Green^®^ Premix Ex Taq™ II (Takara Bio, Inc., Shiga, Japan) and the specified primers (Table 4).

#### 4.6.5. Western Blotting Analysis

The RAW264.7 cells (2 × 10^6^ cells/mL) in a 6-well cell culture plate were subjected to treatment with polysaccharides and LPS for 24 h at 37 °C. Then, the cells were harvested and lysed in radioimmunoprecipitation assay (RIPA) buffer for 30 min before being collected by centrifugation at 12,000× *g* for 10 min at 4 °C. The protein contents were determined using the Micro BCA™ Protein Assay Kit (Thermo Fisher Scientific Inc., Waltham, MA, USA). A total of 30 µg of protein per sample was separated by electrophoresis on 10% SDS-polyacrylamide gel electrophoresis and transferred onto a polyvinylidene fluoride (PVDF) membrane. Primary antibodies against phospho-NF-κB p65, phospho-p38 MAPK, phospho-SAPK/JNK, phospho-p44/42 MAPK (Erk1/2), and alpha-tubulin, along with an anti-rabbit IgG HRP-linked antibody (Abcam, Cambridge, UK), were used to incubate the membrane. The proteins were identified using the Pierce ECL Plus Western Blotting Substrate (Thermo Fisher Scientific Inc., Waltham, MA, USA), and the expression was visualised using the Bio-Rad image analysis program (Bio-Rad, Hercules, CA, USA).

#### 4.6.6. Flow Cytometry Analysis

RAW264.7 cells at a density of 1 × 10^6^ cells/mL in a 6-well cell culture plate treated with the polysaccharides (125–1000 µg/mL) were rinsed with a flow cytometry buffer. Then, 10 µL of anti-CD40-APC (Anti-CD40 (1C10) Allophycocyanin (APC), Thermo Fisher Scientific Inc., Waltham, MA, USA) and 10 µL of anti-CD11b (anti-CD11b (M1/70) fluorescein isothiocyanate (FITC), Thermo Fisher Scientific Inc., Waltham, MA, USA) monoclonal antibodies were stained to each sample and incubated for 30 min at 4 °C in the dark. The unconjugated antibodies were removed by washing the cells with FACS buffer and then resuspended with 1% paraformaldehyde. In order to analyse the flow cytometry, a CytoFLEX Flow Cytometer (Beckman Coulter, Inc., Brea, CA, USA) was used.

### 4.7. Desulphation of Polysaccharide

The desulphation of polysaccharide was based on the method reported by Tarbasa et al. [59]; the JF3 fraction (100 mg) was dissolved in 10 mL of distilled water and eluted with pyridine from a Dowex 50 W resin column, yielding lyophilised polysaccharide–pyridinium salts. Then, the polysaccharide–pyridinium salts were processed at 120 °C and for 40 min to obtain the desulphation sample. The solution was dialysed against distilled water and finally lyophilised to obtain the desulphated JF3. The glycosidic linkage of native JF3 and desulphated JF3 was analysed using GC–MS.

### 4.8. Methylation Analysis of JF3

The samples were methylated according to Ciucanu’s method [60], which was slightly modified. The sample was dissolved in 0.5 mL of dimethyl sulfoxide (DMSO) and methylated with methyl iodide (CH_3_I) and sodium hydroxide powder (NaOH). Partially methylated alditol acetates were created by methylating polysaccharides using acid hydrolysis with 4 M TFA at 100 °C for 6 h. After that, the hydrolysates were decreased with NaBD_4_ and acetylated with acetic anhydride. GC–MS (6890 N/MSD 5973, Agilent Technologies, Santa Clara, CA, USA) was used for the analysis of partly methylated alditol acetates on an HP-5MS capillary column (30 m × 0.25 mm × 0.25 µm) (Agilent Technologies, Santa Clara, CA, USA). The carrier gas was helium, with a constant flow rate of 1.2 mL/min. The oven settings incorporated a temperature program that proceeded from 160 to 210 °C in 10 min and then to 240 °C in 10 min. Thus, a temperature gradient was used at 5 °C/min, the inlet temperature was maintained at 250 °C, and the mass range was set to 35 to 450 m/z. Peaks were assigned by considering retention times and mass spectra.

### 4.9. Statistical Analyses

The experimental design was completed with a completely randomised design (CRD). The analysed data were reported as the mean ± standard deviation (SD) of three independent experiments (*n* = 3) using SPSS version 23.0 software (SPSS, Inc, Chicago, IL, USA) for data analysis. The significance was determined using one-way ANOVA, and Duncan’s new multiple range tests (DMRTs) were used to test the differences between treatment groups (*p* < 0.05).

## 5. Conclusions

The water-soluble polysaccharides isolated from jellyfish (*Lobonema smithii*) were fractionated into three fractions with different sugar, protein, and sulphate contents, as well as different sugar units, exhibiting wide ranges of M_w_ values. The crude and fractionated (JF1, JF2, and JF3) polysaccharides could activate RAW264.7 cells by promoting cell viability and reducing the secretion of NO. In particular, JF3 inhibited the LPS-induced release of factors and cell surface molecules that promote inflammation by decreasing NF-κB and MAPK signalling. Furthermore, JF3 also contributed to a reduction in pro-inflammatory cytokines and an increase in anti-inflammatory cytokines in response to LPS. The increased activity of JF3 could have resulted from its lower molecular weight relative to the crude polysaccharides and the other two fractions. In addition, (1→3)-linked glucopyranosyl, (1→3,6)-linked glucopyranosyl, and (1→3,6)-linked galactopyranosyl residues comprised the main chain of JF3, partially sulfated at O-6 of glucose; furthermore, the other units and terminals could be in the side chains. Therefore, this study suggests that *L. smithii* polysaccharides potentially possess anti-inflammatory activity and could be useful as effective therapeutic agents against inflammation according to the chemical composition and primary molecular structure results, especially for JF3 polysaccharides. To maximise the utilisation of this marine species as a bioactive resource in the pharmaceutical industry, additional research is required on the purification and identification of the active polysaccharides in JF3, and the impact of the functional group and molecular structure on its anti-inflammatory properties should be investigated in future studies.

## Figures and Tables

**Figure 1 marinedrugs-21-00559-f001:**
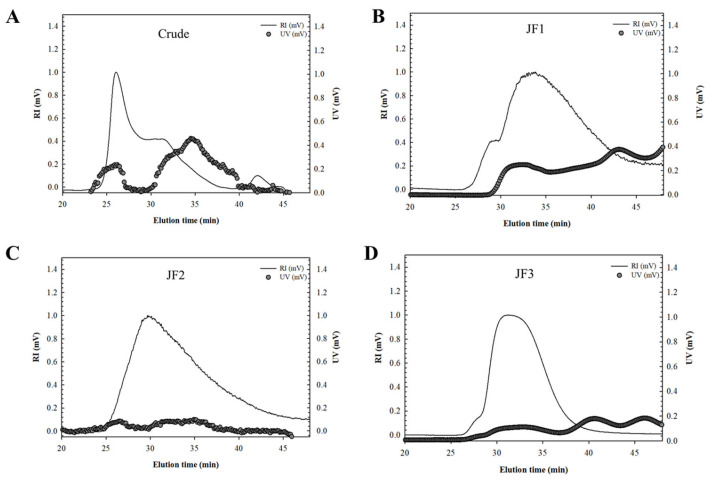
HPSEC chromatograms of crude polysaccharides (**A**) and fractions JF1 (**B**), JF2 (**C**), and JF3 (**D**) isolated from *L. smithii* jellyfish.

**Figure 2 marinedrugs-21-00559-f002:**
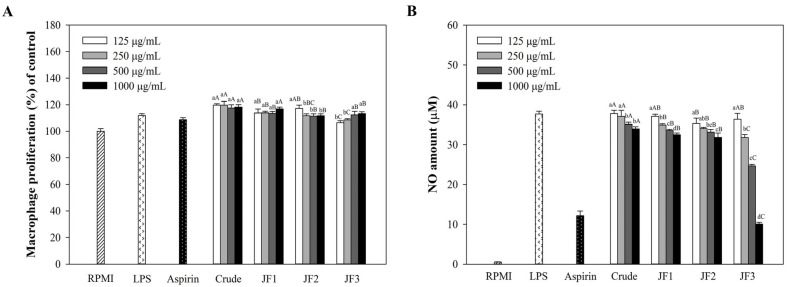
Effect of crude and fractions of *L. smithii* polysaccharides on the viability of RAW264.7 macrophages. The cells (1 × 10^6^ cells/mL) received treatments with varying polysaccharide concentrations (125–1000 μg/mL) and were triggered with LPS (1 μg/mL). The cell viability was determined with a WST assay (**A**). The production of NO was determined with a Griess reagent (**B**). The values are displayed as the mean ± SD (*n* = 3). Different letters ^a,b,c,d^ refer to a significant difference between samples at each concentration, while ^A, B, C^ refer to a significant difference within concentrations at *p* < 0.05.

**Figure 3 marinedrugs-21-00559-f003:**
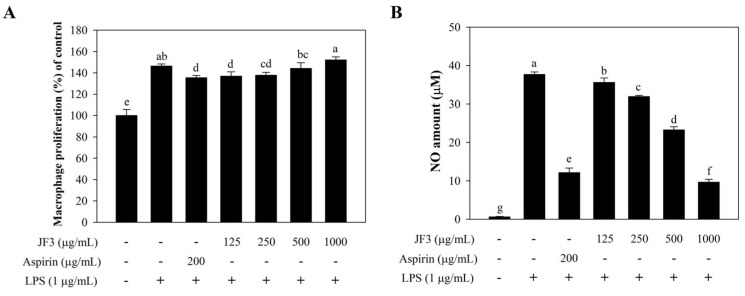
Effect of polysaccharide fraction JF3 on the viability of RAW264.7 macrophages. The cells (1 × 10^6^ cells/mL) received treatments with varying polysaccharide concentrations (125–1000 μg/mL) and were triggered with LPS (1 μg/mL). The cell viability was determined by WST assay (**A**). The production of NO was determined by a Griess reagent (**B**). The values are displayed as the mean ± SD (*n* = 3). The letters ^a, b, c, d, e, f, g^ indicate significant differences (*p* < 0.05) between treatment groups.

**Figure 4 marinedrugs-21-00559-f004:**
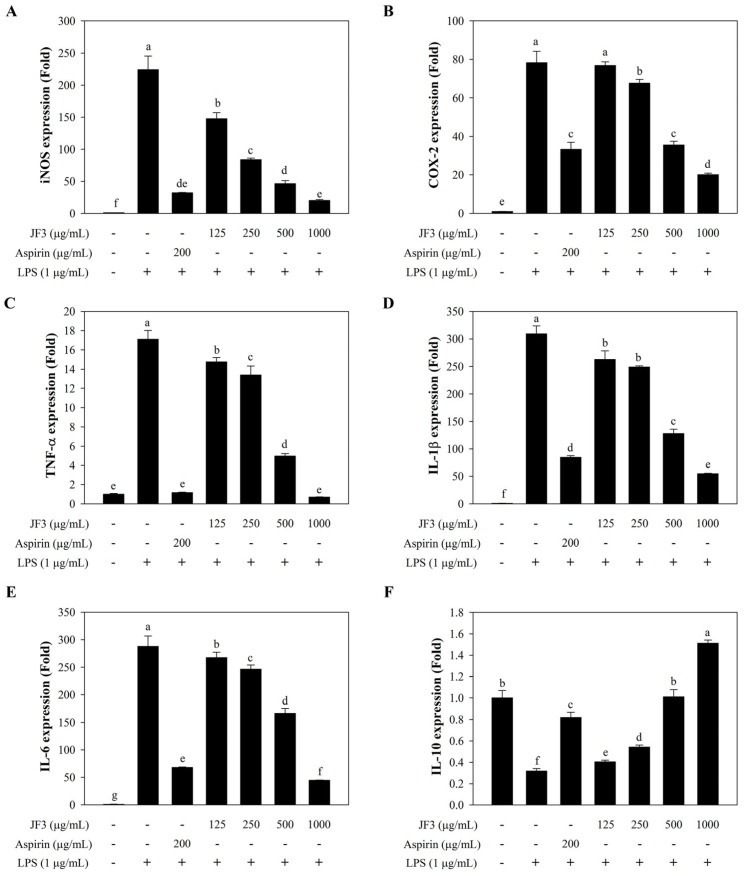
Effects of the JF3 polysaccharide fraction on the secretion of inflammatory mediators and cytokines in RAW264.7 macrophages. The cells (1 × 10^6^ cells/mL) received treatments with varying polysaccharide concentrations (125–1000 μg/mL) and were triggered with LPS (1 μg/mL). The mRNA expressions of iNOS (**A**), COX-2 (**B**), TNF-α (**C**), IL-1β (**D**), IL-6 (**E**), and IL-10 (**F**) were determined by qPCR. The values are displayed as the mean ± SD (*n* = 3). The letters ^a, b, c, d, e, f, g^ indicate significant differences (*p* < 0.05) between treatment groups.

**Figure 5 marinedrugs-21-00559-f005:**
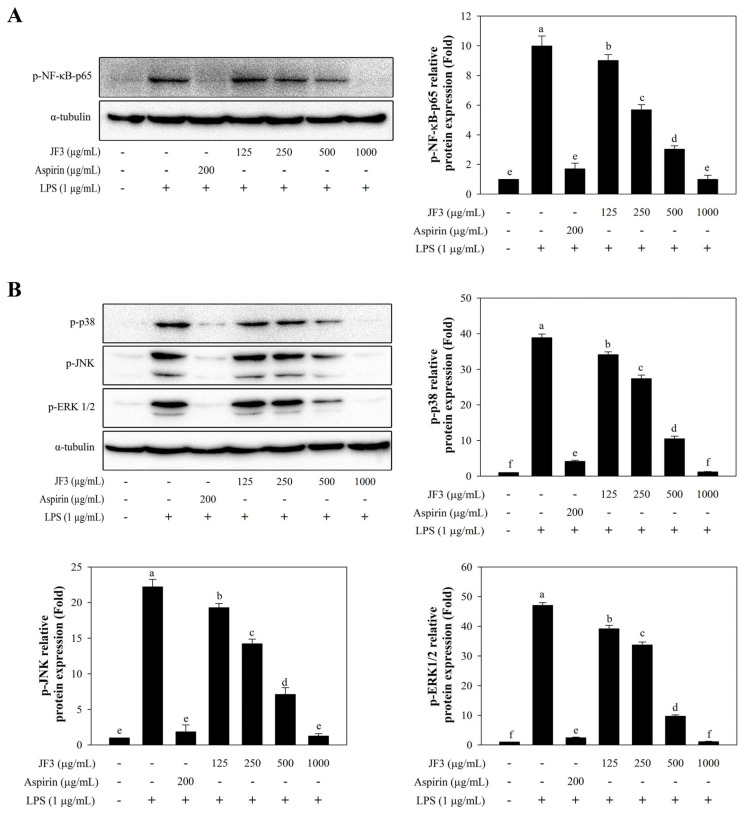
Effects of the JF3 fractionated polysaccharides on the phosphorylated NF-κB subunit 65, ERK1/2, JNK, and p38 MAPK. The cells (2 × 10^6^ cells/mL) received treatments with varying polysaccharide concentrations (125–1000 μg/mL) and were triggered with LPS (1 μg/mL). The levels of protein expression were determined by Western blotting with phospho-NF-κB p65 antibodies (**A**) and specific antibodies to MAPKs (**B**) across three independent experiments. The values are displayed as the mean ± SD (*n* = 3). The letters ^a, b, c, d, e, f^ indicate significant differences (*p* < 0.05) between treatment groups.

**Figure 6 marinedrugs-21-00559-f006:**
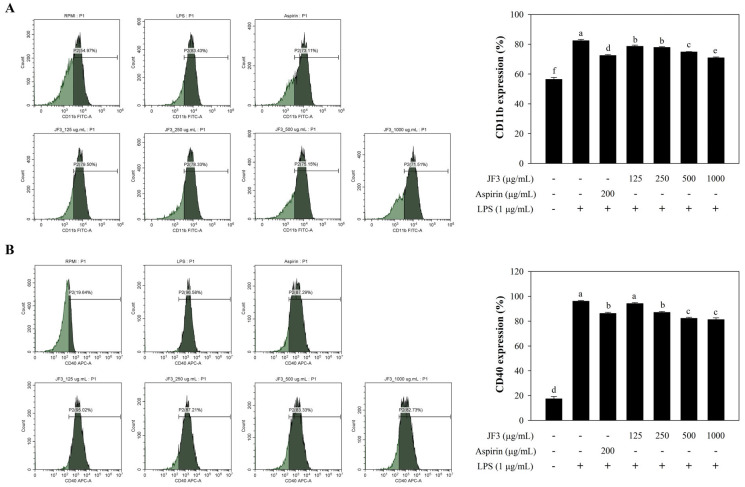
Evaluation of CD11b and CD40 expressions in JF3-treated RAW264.7 macrophages. The cells (2 × 10^6^ cells/mL) received treatments with varying polysaccharide concentrations (125–1000 μg/mL) and were triggered with LPS (1 μg/mL). The expression of CD11b (**A**) and CD40 (**B**) was determined by flow cytometry. The values are displayed as the mean ± SD (*n* = 3). The letters ^a, b, c, d, e, f^ indicate significant differences (*p* < 0.05) between treatment groups.

**Figure 7 marinedrugs-21-00559-f007:**
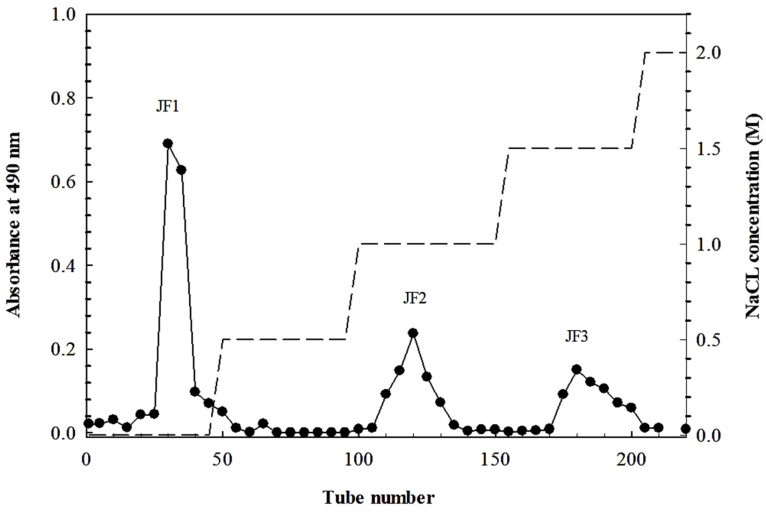
Elution profile of *L. smithii* polysaccharides (JF1, JF2, and JF3) fractionated in a DEAE Sepharose fast flow column.

**Table 1 marinedrugs-21-00559-t001:** Chemical compositions of polysaccharides isolated from *Lobonema smithii*.

Components	*L. smithii* Polysaccharides			
	Crude	JF1	JF2	JF3
Yield (%)	1.30 ± 0.10 ^x^	77.2 ± 0.61 ^y^	13.2 ± 0.22 ^y^	9.70 ± 0.05 ^y^
Total carbohydrate (%)	62.5 ± 0.12 ^a^	72.2 ± 0.06 ^a^	66.0 ± 0.06 ^a^	67.6 ± 0.06 ^a^
Protein (%)	24.1 ± 0.06 ^b^	18.1 ± 0.00 ^b^	13.6 ± 0.04 ^c^	7.12 ± 0.13 ^c^
Sulphate (%)	10.2 ± 0.88 ^c^	8.27 ± 0.21 ^c^	17.3 ± 0.21 ^b^	22.7 ± 0.21 ^b^
Uronic acid (%)	3.17 ± 0.06 ^d^	1.47 ± 0.03 ^d^	2.97 ± 0.02 ^d^	3.40 ± 0.10 ^d^
Monosaccharide content (%)				
D-galactose	26.2 ± 0.20 ^b^	41.0 ± 1.00 ^a^	12.5 ± 0.02 ^c^	28.4 ± 0.00 ^b^
D-glucose	33.1 ± 0.10 ^a^	40.2 ± 0.21 ^b^	52.4 ± 0.36 ^a^	56.7 ± 0.05 ^a^
L-arabinose	15.0 ± 0.00 ^c^	11.7 ± 0.07 ^c^	15.4 ± 0.15 ^b^	0.60 ± 0.15 ^d^
D-mannose	15.2 ± 0.10 ^c^	3.83 ± 0.06 ^d^	10.6 ± 0.11 ^d^	13.7 ± 0.06 ^c^
L-rhamnose	8.30 ± 0.20 ^d^	1.80 ± 0.03 ^e^	6.36 ± 0.15 ^e^	0.54 ± 0.03 ^d^
L-fucose	2.23± 0.25 ^e^	1.30 ± 0.06 ^e^	2.53 ± 0.06 ^f^	0.10 ± 0.10 ^e^

^x^ Yield, (the amount weight of crude/powdered sample) × 100. ^y^ Yield, (the amount weight of fractions/crude injected into column) × 100. % dry basis. Different letters ^a,b,c,d,e,f^ indicate significant differences (*p* < 0.05) between the groups in each column.

**Table 2 marinedrugs-21-00559-t002:** Molecular weight of polysaccharides isolated from *Lobonema smithii*.

*L. smithii* Polysaccharides	M_w_ (kDa)	R_g_ (nm)	SV_g_ (cm^3^/g)
Crude	865.0 ± 8.71 ^a^	124.0 ± 3.90 ^a^	5.57 ± 0.57 ^a^
JF1	477.6 ± 7.94 ^c^	77.7 ± 3.61 ^c^	2.49 ± 0.39 ^c^
JF2	524.1 ± 7.55 ^b^	94.4 ± 2.53 ^b^	4.05 ± 0.26 ^b^
JF3	293.0 ± 6.63 ^d^	56.3 ± 7.20 ^d^	1.59 ± 0.66 ^d^

Different letters ^a,b,c,d^ refer to significant differences (*p* < 0.05) between the groups in each column.

**Table 3 marinedrugs-21-00559-t003:** Glycosidic linkage analysis of JF3.

Characteristic Fragment Ions (m/z)	Methylation Product	Glycosidic Linkage	JF3 (%)	D-JF3 (%)
84, 102, 118, 129, 162, 207	1,5-di-O-acetyl-2,3,4,6-tetra-O-methyl-Man	Man*p*-(1→	9.1	10.5
87, 101, 118, 129, 161, 206, 234	1,3,5-tri-O-acetyl-2,4,6-tri-O-methyl-Glc	→3)-Glc*p*-(1→	16.4	48.4
87, 101, 118, 129, 162, 189, 234	1,5,6-tri-O-acetyl-2,3,4-tri-O-methyl-Gal	→6)-Gal*p*-(1→	1.00	1.30
87, 101, 118, 129, 189, 234	1,3,5,6-tetra-O-acetyl-2,4-di-O-methyl-Glc	→3,6)-Glc*p*-(1→	45.2	13.5
87, 101, 118, 129, 189, 234	1,3,5,6-tri-O-acetyl-2,4-di-O-methyl-Gal	→3,6)-Gal*p*-(1→	28.3	26.3

**Table 4 marinedrugs-21-00559-t004:** The sequences of primers used in real-time PCR.

Target Genes	Sequences of the Primers (5′ to 3′)	
	Forward	Reverse
IL-1β	GGGCCTCAAAGGAAAGAATC	TACCAGTTGGGGAACTCTGC
IL-6	AGTTGCCTTCTTGGGACTGA	CAGAATTGCCATTGCACAAC
IL-10	TACCTGGTAGAAGTGATGCC	CATCATGTATGCTTCTATGC
TNF-α	ATGAGCACAGAAAGCATGATC	TACAGGCTTGTCACTCGAATT
iNOS	TTCCAGAATCCCTGGACAAG	TGGTCAAACTCTTGGGGTTC
COX-2	AGAAGGAAATGGCTGCAGAA	GCTCGGCTTCCAGTATTGAG
β-actin	CCACAGCTGAGAGGGAAATC	AAGGAAGGCTGGAAAAGAGC

## Data Availability

The datasets used and/or analysed during the current study are available from the corresponding author upon reasonable request.

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
