# Peer review of "In Vitro Anti-Inflammatory Activity and Structural Characteristics of Polysaccharides Extracted from Lobonema smithii Jellyfish"

_marinedrugs, 2023, doi:10.3390/md21110559_

Round 1
Reviewer 1 Report
Comments and Suggestions for Authors
This paper investigates the crude polysaccharides were extracted from white jellyfish (Lobonema smithii) using water extraction and fractionated using an ion-exchange chromatography to obtain three different fractions (JF1, JF2, and JF3). The chemical characteristics of the four polysaccharides were investigated, alongside their anti-inflammatory effect in LPS-stimulated RAW264.7 cells.To my opinion, there is still much to be said in detail in this article. Therefore, it can be considered for publication after careful corrections.
My comments to improve their listed below:
1. The structure of the article needs to be adjusted and it is suggested that part 2.9 be placed before 2.3. Also, did the authors modify JF1 and JF2 with methylation, and are they structurally different from JF3 specifically?
2. The authors determined the molecular weight of the polysaccharide, what about the polydispersity index (PDI) of the polysaccharide? Is there any specific data? Please add a description.
3. What can the radius of rotation (Rg) of a polysaccharide peak tell us? What is the specific significance, and what does the fact that the three polysaccharides have different Rg indicate? Please add discussion.
4. Line 314, I'm curious as to why the samples were left for two months before extracting the polysaccharide polysaccharides, is there any special significance to this?
5. In section 4.2, what is the specific solid-liquid ratio for polysaccharide extraction? Please add a description.
6. In the discussion, please add a comparison of the polysaccharide structure with results from other references. What are the differences?
7. The authors determined the composition, molecular weight, and other structural information of the polysaccharide, so is there any specific relationship between activity and structure? Please add to the discussion.
Comments on the Quality of English Language
It is ok
Author Response
- The structure of the article needs to be adjusted and it is suggested that part 2.9 be placed before 2.3. Also, did the authors modify JF1 and JF2 with methylation, and are they structurally different from JF3 specifically?
- Thank you very much.
- In this study, we studied the structure (Linkage analysis) of the most anti-inflammatory JF3 polysaccharides, so we add the methylation analysis part after the anti-inflammatory activity.
- For the JF1 and JF2 on methylation analysis would be taken into consideration as a further work. Further study on the fine structures of fractions and biological activities provides for a greater understanding of the deeper correlation between structures and biological processes to accomplish their maximum effects.
- The authors determined the molecular weight of the polysaccharide, what about the polydispersity index (PDI) of the polysaccharide? Is there any specific data? Please add a description.
- In terms of the polydispersity index (PDI), we did not determine, however, we provide the specific volume for gyration (SVg) instead which shows the density of the polysaccharides relates to the molecular compactness.
- We add more data on specific volume for gyration (SVg): The SVg is inversely related to the degree of molecular compactness, giving the theoretical gyration volume per unit of molar mass and providing mass-based information on polysaccharide density
- Page 3-4, Line 139-157: We add sentence “The individual volume for gyration (SVg) of the SPs may be derived from the values of Mw and Rg using the following equation, as described by You and Lim [28].
where the values for SVg, Mw, and Rg are cm3/g, g/mol and nm, respectively, and N is Avogadro’s number (6.02×1023/mol). The SVg is inversely related to the degree of molecular compactness, giving the theoretical gyration volume per unit of molar mass and providing mass-based information on polysaccharide density [28]. The overall SVg values of the crude and fractionated polysaccharides are shown in Table 2. The SVg values for the crude polysaccharides were 5.57 cm3/g, and its fractions were 5.57, 2.49, 4.05, and 1.59, respectively. The crude SVg values were significantly higher than those of the fractions. This finding indicated that the crude possessed less compact and more extended conformational structures than those of JF2, following JF1 and JF3, which are the most compact. These discrepancies in Mw, Rg, and SVg were most likely caused by a combination of factors, including the extraction and purification processes, as well as the analytical methods used in each investigation.
- We have changed the Table 2
Table 2. Chemical compositions of polysaccharides isolated from Lobonema smithii.
L. smithii polysaccharides |
Mw (kDa) |
Rg (nm) |
SVg (cm3/g) |
Crude |
865.0±8.71a |
124.0±3.90a |
5.57±0.57a |
JF1 |
477.6±7.94c |
77.7±3.61c |
2.49±0.39c |
JF2 |
524.1±7.55b |
94.4±2.53b |
4.05±0.26b |
JF3 |
293.0±6.63d |
56.3±7.20d |
1.59±0.66d |
Different letters a,b,c,d,e refer to significant differences (p < 0.05) between the groups in each column.
-
- You SG, Lim ST. (2000) Molecular Characterization of Corn Starch Using an Aqueous HPSECMALLS-RI System Under Various Dissolution and Analytical ConditionsCereal Chem., 77, 303-308.
- What can the radius of rotation (Rg) of a polysaccharide peak tell us? What is the specific significance, and what does the fact that the three polysaccharides have different Rg indicate? Please add discussion.
- Page 3-4, Line 133-138: We revised from “The Rg values were 77.3 and 94.4 nm for the JF1 and JF2 fraction and 56.3 nm for fraction JF3.” to “Rg is explained as the distribution of units of a polysaccharide around its axis and can define the size of the polymer, which can be calculated from light scattering up to various angles of the MALLS system. Figure 2 shows that the Rg values were 124, 77.3, and 94.4 nm for the crude, JF1, and JF2 fractions and 56.3 nm for fraction JF3, indicating that in this result, the size of the polymer is in accordance with the Mw.”
- Line 314, I'm curious as to why the samples were left for two months before extracting the polysaccharide polysaccharides, is there any special significance to this?
- No significance to the time (2 months) of the sample before extraction.
- Page 11, Line 386-387: This sentence could be revised from "The jellyfish was packed into a polyethylene bag, kept in a freezer at −20°C, and used for polysaccharide extraction after two months." to "The jellyfish was placed into a polyethylene bag, frozen at −20°C, and used for polysaccharide extraction."
- In section 4.2, what is the specific solid-liquid ratio for polysaccharide extraction? Please add a description.
- In section 4.2, Page 11, Line 396-399: We revised the sentence “All water extracts were combined and precipitated with 98% (v/v) ethanol, which was then subjected to filtration.” to “All water extracts were combined and precipitated from the supernatant using four volumes of cold 98% (v/v) ethanol and stored overnight at 4°C, which was then subjected to filtration.”
- In the discussion, please add a comparison of the polysaccharide structure with results from other references. What are the differences?
- Page 9, Line 285-311 and Page 11, Line 357-376: We added more discussion on
polysaccharide structure with results from other references.
- Thank you for your suggestion.
- The authors determined the composition, molecular weight, and other structural information of the polysaccharide, so is there any specific relationship between activity and structure? Please add to the discussion.
- Page 11, Line 357-376: We added more discussion about relationship between structure and bioactivity.
- Thank you for your suggestion.

Reviewer 2 Report
Comments and Suggestions for Authors
This paper present data on the anti-inflammatory properties and structural characteristics of polysaccharides from the jellyfish Lobonema smithii. My comments concern my area of expertise, the polysaccharide composition and structure analysis; I have not commented on the anti-inflammatory data.
In the introduction, the authors start by stating that marine polysaccharides have different biological properties, but the rest of the paragraph concerns lipopolysaccharide and inflammation rather than expanding on the biological effects of marine polysaccharides. This should be addressed.
The second paragraph discusses the anti-inflammatory properties of jellyfish polysaccharides, but does not discuss what is already known about the polysaccharide compositions and structures. This needs to be included.
At line 64 the authors state that information on the anti-inflammatory properties of L. smithii polysaccharides is limited. Is there any information at all? If not then state that.
The authors report the monosaccharide composition of the crude polysaccharide extract and three fractions, as well as the linkage composition of most anti-inflammatory fraction, JF3. However, discussion in relation to the literature is limited to one sentence at line 253. In section 3, Discussion, there is nothing concerning the polysaccharide structure in relation to published data on jellyfish polysaccharide structure or biological properties.
The methylation analysis data present a major concern which needs to be addressed. The monosaccharide data for JF3 shows abundances of monosaccharides in decreasing amounts: Man (~45%), Gal (~28%), Glc (~15%), Fuc (~8%), Rha (~2%), Ara (~1%), but methylation shows Glc linkages most abundant ~45%, then Gal (~30%), Man (~18%) and Ara (~8%); Fuc and Rha linkages are absent. This represents some very large discrepancies that must be addressed. Further, there are several anomalies in the linkage data shown in Table 3. Firstly, the methylation product at 6.4 min is not that for T-Araf, but of T-Arap. In addition, it is very unlikely that the methylation product for 3,6-Glc (11.8 min) would elute before those for 3-Gal (12.1 min) and 2-Man (12.4 min). Why are methylation products for Fuc or Rha linkages not detected when these sugars are more abundant than Ara? It is also curious that the terminal/branch point ratio is close to 1:1; this would not be expected for sulfated polysaccharides unless they were desulfated before linkage analysis. In order for this data to be assessed properly, please supply additional information, specifically a total ion chromatogram showing the elution of the derivatives and mass spectra for each of the peaks with interpretations of how the linkages were deduced.
Line 127: the molecular weight data is shown in Table 2, not Table 3.
Lines 130 and 131: the molecular weights reported for publications [20] and [12] are incorrect; the molecular weight of RP-JSP2 is 590 kDa (not 250 kDa) and that for JSP1 (JSP11 in Li et al) is 1250 kDa and not 12.5 kDa.
Line 132: the most obvious reason for differences in molecular weight is surely the species of jellyfish used?
Line 230: this should be 2.8.
Line 253: how is this similar to that reported, when there are no uronic acid linkages, no 3,6-Man and no 6-Gal? This is a very different structure.
Line 306: how was the identity of the jellyfish confirmed?
Line 324: were the extracts precipitated with 98% ethanol, or was 98% ethanol added to give a final concentration of ethanol of, for example, 80%. Please be more specific.
Line 347: at what temperature and for how long were samples hydrolysed?
Line 348: please be more specific; how were samples reduced to alditols and how were they acetylated?
Line 357: what column/s was used, what eluent, column temperature, flow rate, etc? What was the refractive index refractive index increments used for the MALLS to determine molecular weight and size?
Line 412: please rewrite this sentence to make sense.
Line 415: please provide more details of the GC and MS conditions used. How did you identify and the methylation products and deduce their linkages?
Comments on the Quality of English Language
I strongly suggest that the authors have their manuscript edited by an English language expert to improve the language and grammar throughout their paper. For instance, the first sentence of the introduction is repetitive. There are similar examples of areas where improvements need to be made.
Author Response
In the introduction, the authors start by stating that marine polysaccharides have different biological properties, but the rest of the paragraph concerns lipopolysaccharide and inflammation rather than expanding on the biological effects of marine polysaccharides. This should be addressed.
- It was corrected in lines 38-41.
The second paragraph discusses the anti-inflammatory properties of jellyfish polysaccharides, but does not discuss what is already known about the polysaccharide compositions and structures. This needs to be included.
- It was corrected.
At line 64 the authors state that information on the anti-inflammatory properties of L. smithii polysaccharides is limited. Is there any information at all? If not then state that.
- It was corrected lines 66-68.
The authors report the monosaccharide composition of the crude polysaccharide extract and three fractions, as well as the linkage composition of most anti-inflammatory fraction, JF3. However, discussion in relation to the literature is limited to one sentence at line 253. In section 3, Discussion, there is nothing concerning the polysaccharide structure in relation to published data on jellyfish polysaccharide structure or biological properties.
The methylation analysis data present a major concern which needs to be addressed. The monosaccharide data for JF3 shows abundances of monosaccharides in decreasing amounts: Man (~45%), Gal (~28%), Glc (~15%), Fuc (~8%), Rha (~2%), Ara (~1%), but methylation shows Glc linkages most abundant ~45%, then Gal (~30%), Man (~18%) and Ara (~8%); Fuc and Rha linkages are absent. This represents some very large discrepancies that must be addressed. Further, there are several anomalies in the linkage data shown in Table 3. Firstly, the methylation product at 6.4 min is not that for T-Araf, but of T-Arap. In addition, it is very unlikely that the methylation product for 3,6-Glc (11.8 min) would elute before those for 3-Gal (12.1 min) and 2-Man (12.4 min). Why are methylation products for Fuc or Rha linkages not detected when these sugars are more abundant than Ara? It is also curious that the terminal/branch point ratio is close to 1:1; this would not be expected for sulphated polysaccharides unless they were desulphated before linkage analysis. In order for this data to be assessed properly, please supply additional information, specifically a total ion chromatogram showing the elution of the derivatives and mass spectra for each of the peaks with interpretations of how the linkages were deduced.
- Thank you very much for the insight review of the linkage analysis. As the reviewer mentioned a very important point about the structural analysis, we re-assessed the data properly and carefully. First, the monosaccharide composition was mistaken by adding the different monosaccharides with different percentages. We have carefully checked and revised the monosaccharide composition of JF3 as shown in Table 1, Glc (44.5%), Gal (29.6%), Man (17.4%), Fuc (1.13%), Rha (0.34%), Ara (8.50%) with consistency with the methylation analysis.
Components |
L. smithii polysaccharides |
|||
Crude |
JF1 |
JF2 |
JF3 |
|
Yield (%) |
1.30±0.10x |
77.2±0.61y |
13.2±0.22y |
9.70±0.05y |
Total carbohydrate (%) |
62.5±0.12a |
72.2±0.06a |
66.0±0.06a |
73.6±0.06a |
Protein (%) |
24.1±0.06b |
18.1±0.00b |
13.6±0.04c |
7.12±0.13c |
Sulphate (%) |
10.2±0.88c |
8.27±0.21c |
17.3±0.21b |
15.7±0.21b |
Uronic acid (%) |
3.17±0.06d |
1.47±0.03d |
2.97±0.02 d |
3.40±0.10d |
Monosaccharide content (%) |
||||
D-Galatose |
26.2±0.20b |
41.0±1.00a |
12.5±0.02c |
29.6±0.20b |
D-Glucose |
33.1±0.10a |
40.2±0.21b |
52.4±0.36a |
44.5±0.50a |
L-Arabinose |
15.0±0.00c |
11.7±0.07c |
15.4±0.15b |
8.50±0.15d |
D- Mannose |
15.2±0.10c |
3.83±0.06d |
10.6±0.11d |
17.4±0.06c |
L-Rhamnose |
8.30±0.20d |
1.80±0.03e |
6.36±0.15e |
0.34±0.15f |
L-Fucose |
2.23± 0.25e |
1.30±0.06e |
2.53±0.06f |
1.13±0.14e |
- We have revised the T-Araf, to T-Arap in Table 3.
- Table 3, We have revised and added more information on the methylation of JF3 and desulphated JF3 to specify the position of sulphated connection.
- We revised the part Methylation Analysis of JF3:
The glycosidic linkage of the most anti-inflammatory JF3 polysaccharides was analysed by GC–MS. In Table 3, the alditol acetate products of JF3 mainly consisted of 2,4,6-Me3-Glcp, 2,4,6-Me3-Galp, and 3,4,6-Me3-Manp, indicating the presence of (1→3)-linked glucopyranosyl, (1→3)-linked galactopyranosyl, and (1→2)-linked mannopyranosyl residues. In addition, JF3 also included the amounts of 2,4-Me2-Glcp, 3,6-Me2-Manp implying the presence of (1→3,6)-linked glucopyranosyl, (1→2,4)-linked mannopyranosyl residues. Some branches in the backbone observed by the terminal were correlated with arabinose and galactose. After desulphated the fraction JF3 by the solvolytic desulphation method, the methylation analysed of desulphated-JF3 exhibited the decrease in 2,4-di-O-methyl-glucitol acetate and 3,6-di-O-methyl-mannitol acetate with a concomitant increase in 2,4,6-tri-O-methyl-glucitol acetate and 3,4,6-tri-O-methyl-mannitol acetate with no changes in the proportions of other methylated alditol acetates (Table 3). This suggested that the sulphate groups were mostly linked at the O-6 position of the (1→3,6)-linked glucopyranosyl and at the O-4 position of the (1→2,4)-linked mannopyranosyl residues.
- We have changed the Table 3. Glycosidic linkage analysis of JF3
Characteristic fragment ions (m/z) |
Methylation product |
Glycosidic linkage |
JF3 (%) |
De-Sulphate JF3 (%) |
89, 101, 117, 131,161,206 |
1,5-di-O-acetyl-2,3,4-tri-O-methyl-Ara |
Arap-(1→ |
8.5 |
8.1 |
89, 101, 117, 131,145, 161,206 |
1,5-di-O-acetyl-2,3,4,6-tretra-O-methyl-Gal |
Galp-(1→ |
9.8 |
9.5 |
89, 101, 117, 131,145, 161,234 |
1,3,5-tri-O-acetyl-2,4.6-tri-O-methyl-Glc |
→3)-Glcp-(1→ |
23.7 |
32.6 |
89, 101, 117, 131,161, 189,234 |
1,3,5,6-tretra-O-acetyl-2,4-di-O-methyl-Glc |
→3,6)-Glcp-(1→ |
21.5 |
12 |
87, 89, 117, 131,161, 203,234, 277 |
1,3,5-tri-O-acetyl-2,4,6-tri-O-methyl-Gal |
→3)-Galp-(1→ |
19.1 |
20.2 |
87, 89, 131,161, 190,206,234 |
1,2,5-tri-O-acetyl-3,4,6-tri-O-methyl-Man |
→2)- Manp-(1→ |
5.6 |
12.5 |
87, 89,99,113, 131,190,234 |
1,2,4,5-tretra-O-acetyl-3,6-di-O-methyl-Man |
→2,4)-Manp-(1→ |
11.8 |
5.1 |
- Thank you again for your precious comment.
Line 127: the molecular weight data is shown in Table 2, not Table 3.
- It was corrected.
Lines 130 and 131: the molecular weights reported for publications [20] and [12] are incorrect; the molecular weight of RP-JSP2 is 590 kDa (not 250 kDa) and that for JSP1 (JSP11 in Li et al) is 1250 kDa and not 12.5 kDa.
- It was corrected.
Line 132: the most obvious reason for differences in molecular weight is surely the species of jellyfish used?
- I think the variation in the molecular weight appeared to be significantly impacted by many factors such as the extraction processes, type, species, etc.
Line 230: this should be 2.8.
- It was corrected.
Line 253: how is this similar to that reported, when there are no uronic acid linkages, no 3,6-Man and no 6-Gal? This is a very different structure.
- Page 8, Line 279-282: We have changed the sentence “Li, et al. (2017) indicated that the linear linkage of the fractionated polysaccharides of jellyfish (JSP-11) which was previously reported as a (1→3,6)-Manp and (1→6)-Galp and (1→)-GlcpA as terminal [7].”
Line 306: how was the identity of the jellyfish confirmed?
- The jellyfish was obtained from local fisheries in La-ngu District, Satun Province, Thailand. There are the main areas to catch and pre-processing white jellyfish in the southern part of Thailand. Moreover, the expert from the southern part of Thailand also confirmed the identity of this white jellyfish.
Line 324: were the extracts precipitated with 98% ethanol, or was 98% ethanol added to give a final concentration of ethanol of, for example, 80%. Please be more specific.
- In section 4.2, Page 11 Line 396-399: We revised the sentence “All water extracts were combined and precipitated with 98% (v/v) ethanol, which was then subjected to filtration.” to “All water extracts were combined and precipitated from the supernatant using four volumes of cold 98% (v/v) ethanol and stored overnight at 4°C, which was then subjected to filtration.”
Line 347: at what temperature and for how long were samples hydrolysed?
- Line 420-426: We revised from “The sample was hydrolysed with trifluoroacetic acid (TFA, 4M) and modified into alditol acetates with acetic anhydride” to “The sample was hydrolysed with trifluoroacetic acid (TFA, 4M) at 100°C for 6 h, then reduced in water with sodium borodeuteride (NaBD4) and acetylated with acetic anhydride.”
Line 348: please be more specific; how were samples reduced to alditols and how were they acetylated?
- Line 420-426: We add more specific detail and revised from We revised from “The sample was hydrolysed with trifluoroacetic acid (TFA, 4M) and modified into alditol acetates with acetic anhydride” to “The sample was hydrolysed with trifluoroacetic acid (TFA, 4M) at 100°C for 6 h, then reduced in water with sodium borodeuteride (NaBD4) and acetylated with acetic anhydride.”
Line 357: what column/s was used, what eluent, column temperature, flow rate, etc? What was the refractive index refractive index increments used for the MALLS to determine molecular weight and size?
- Line 428-437: We add more detail of measurement of molecular weights “The Mw of the four polysaccharides was estimated using the HPSEC-UVMALLS-RI system), which included a pump (Waters 510, Milford, MA, USA), an injector valve with a 200 μL sample loop (model 7072, Rheodyne), SEC columns (TSK G5000 PW, 7.5 mm × 600 mm; TosoBiosep, Mongomeryville, PA, USA), a UV detector at 280 nm (Waters 2487), a multi-angle laser light scattering detector (HELEOS, Wyatt Technology Corp, Santa Barbara, CA, USA), and a refractive index detector (Waters 2414). At a flow rate of 0.4 mL/min, the mobile phase of this system contained 0.15 M NaNO3 and 0.02% NaN3. Each sample was immersed in DW, and then heated for 15 min at 75°C before being injected into MALLS. The Mw and Rg were estimated using the ASTRA version 6.0 software (Wyatt Technology Corp.).
Line 412: please rewrite this sentence to make sense.
- We rewrite the sentence from “The best activity sample was dissolved in DMSO and methylated with CH3I and NaOH powder.” to “The sample was dissolved in 0.5 mL dimethyl sulfoxide (DMSO) and methylated with methyl iodide (CH3I) and sodium hydroxide powder (NaOH). Partially methylated alditol acetates were created by methylating polysaccharides using acid hydrolysis with 4 M TFA at 100°C for 6 h. After that, the hydrolysates were decreased with NaBD4, and acetylated with acetic anhydride.”
Line 415: please provide more details of the GC and MS conditions used. How did you identify and the methylation products and deduce their linkages?
- Page 14, Line 503-510: We added the detail of the GC and MS conditions “The GC-MS (6890N/MSD 5973, Agilent Technologies, Santa Clara, CA) was used for the analysis of partly methylated alditol acetates on a HP-5MS capillary column (30 m × 0.25 mm × 0.25 m) (Agilent Technologies, Santa Clara, CA, USA). The carrier gas was helium, with a constant flow rate of 1.2 mL/min. The oven settings incorporated a temperature program that proceeded from 160 to 210°C in 10 min and then to 240°C in 10 min. Thus, a temperature gradient was used at 5°C/min, the inlet temperature maintained at 250°C, and the mass range was set to 35 to 450 m/z. Peaks were assigned by considering retention times and mass spectra.”
Comments on the Quality of English Language
I strongly suggest that the authors have their manuscript edited by an English language expert to improve the language and grammar throughout their paper. For instance, the first sentence of the introduction is repetitive. There are similar examples of areas where improvements need to be made.
- We did English proofreading as an attached certificate file.

Reviewer 3 Report
Comments and Suggestions for Authors
The work represents an original investigation of the isolation, characterization, and evaluation of the in vitro anti-inflammatory activity of three fractions of polysaccharides obtained from Lobonema smithii (White jellyfish).
The polysaccharides were characterized and evaluated the anti-inflammatory activity following known procedures and the appropriate techniques were used for this purpose, which was duly cited.
One of the fractions presented relevant activity, compared to aspirin.
It is only recommended to enlarge the images of Figures 4, 5, and 6 so that they can be appreciated better and taking into account that in the possible final version they will still have to be reduced.
Based on the above, I consider that this work has sufficient quality and relevance for publication in Marine drugs.
Author Response
Comments and Suggestions for Authors:
The work represents an original investigation of the isolation, characterization, and evaluation of the in vitro anti-inflammatory activity of three fractions of polysaccharides obtained from Lobonema smithii (White jellyfish).
The polysaccharides were characterized and evaluated the anti-inflammatory activity following known procedures and the appropriate techniques were used for this purpose, which was duly cited. One of the fractions presented relevant activity, compared to aspirin.
It is only recommended to enlarge the images of Figures 4, 5, and 6 so that they can be appreciated better and taking into account that in the possible final version they will still have to be reduced.
Based on the above, I consider that this work has sufficient quality and relevance for publication in Marine drugs.
- Thank you for your time to review this manuscript.
- Regarding reviewer 3, it was suggested to enlarge the images of Figures 4, 5, and 6.
- We revised it as per the suggestion. Thank you very much.

Reviewer 4 Report
Comments and Suggestions for Authors
This article focuses on the structural analysis and anti-inflammatory activity of jellyfish polysaccharides. The monosaccharide composition and molecular weight of the crude polysaccharide and purified fractions were determined, as well as methylation analysis of the most active fraction.
However, the exact structure of this polysaccharide has not been established, and additional analyzes are needed to complete the structural analysis of the most active fraction. The issues for revision are summarized below.
1. The introduction should be expanded with additional information on the bioactive polysaccharides of jellyfish previously described in the literature. The results obtained should also be compared with the results obtained for jellyfish polysaccharides.
2. It is unclear about inter-connection of monosaccharide units and their anomeric configuration. In addition to uronic acids, they may also contain N-acetylated amino sugars. In order to complete the structural analysis, NMR spectra should be measured and interpreted. FTIR spectra could help determine the location of sulfate groups as well as the presence of carboxyl and amide groups.
3. Discussion should be expanded by the structure - activity relationship based on the results obtained for the purified fractions.
4. Table 2: these are molecular weight/size parameters, not chemical composition!
5. Line 365: should be "(1 → 3)- linked α-D-glucopyranosyl"
Comments on the Quality of English Language
Minor editing of Englishgrammar and style is required.
Author Response
Comments and Suggestions for Authors:
This article focuses on the structural analysis and anti-inflammatory activity of jellyfish polysaccharides. The monosaccharide composition and molecular weight of the crude polysaccharide and purified fractions were determined, as well as methylation analysis of the most active fraction.
However, the exact structure of this polysaccharide has not been established, and additional analyzes are needed to complete the structural analysis of the most active fraction. The issues for revision are summarized below.
- The introduction should be expanded with additional information on the bioactive polysaccharides of jellyfish previously described in the literature. The results obtained should also be compared with the results obtained for jellyfish polysaccharides.
- Lines 59–61: The introduction provides information on the bioactive polysaccharides of jellyfish.
- In addition, our results were compared with the results from other jellyfish polysaccharides, as shown in lines 131–134, 278–280.
- Thank you for the precious comments.
- It is unclear about inter-connection of monosaccharide units and their anomeric configuration. In addition to uronic acids, they may also contain N-acetylated amino sugars. In order to complete the structural analysis, NMR spectra should be measured and interpreted. FTIR spectra could help determine the location of sulfate groups as well as the presence of carboxyl and amide groups.
- Regarding the suggestion made by reviewer 4 to clarify the anomeric configuration and uronic acids as they may also contain N-acetylated amino sugars by the analysis of FT-IR and NMR.
- In this manuscript, we provide the data on desulphation to obtain the location of sulfate groups, as shown in Table 3. However, to the best of our knowledge, this manuscript already includes sufficient data on the primary structure of polysaccharides extracted from Lobonema smithii. The FT-IR analysis, which is an important experiment, provides information on the location of sulfate groups as well as the presence of carboxyl and amide groups. Moreover, NMR analysis will exhibit the type of anomeric configuration.
- Therefore, the next experiment will investigate the effect of the functional group, which is determined by FT-IR, molecular weight, and fine structure (1D and 2D-NMR), on the anti-inflammatory properties of JF3 polysaccharides from jellyfish ( smithii) in order to elucidate their precise impact on anti-inflammatory activity.
- Discussion should be expanded by the structure - activity relationship based on the results obtained for the purified fractions.
- We revised the discussion on the structure-activity relationship based on the advice.
- Table 2: these are molecular weight/size parameters, not chemical composition!
- We revised the title of Table 2: Molecular weight of polysaccharides isolated from Lobonema smithii.
- Line 365: should be "(1 → 3)- linked α-D-glucopyranosyl"
- We revised (1 → 3)- linked α-D-glucopyranosyl.
- Thank you for your time to review this manuscript.

Round 2
Reviewer 1 Report
Comments and Suggestions for Authors
The authors have revised the manuscript according to the reviewers. It can be accepted now
Comments on the Quality of English Language
The authors have revised the manuscript according to the reviewers. It can be accepted now
Author Response
The authors have revised the manuscript according to the reviewers. It can be accepted now
- It was corrected.
- Thanks.
Comments on the Quality of English Language
The authors have revised the manuscript according to the reviewers. It can be accepted now
- It was corrected.
- Thanks.

Reviewer 2 Report
Comments and Suggestions for Authors
In my first review I asked the authors to supply a total ion chromatogram showing the elution of the derivatives and mass spectra for each of the peaks with interpretations of how the linkages were deduced. This is necessary as there are anomalies in their data that require investigation: 1) the elution order of the derivatives they identify is highly unlikely, and 2) the mass fragments they provide in their revised Table 3 is inconsistent with what would be expected from the methods provided in section 4.8.
Comments on the Quality of English Language
Minor editing required
Author Response
Comments and Suggestions for Authors
In my first review I asked the authors to supply a total ion chromatogram showing the elution of the derivatives and mass spectra for each of the peaks with interpretations of how the linkages were deduced. This is necessary as there are anomalies in their data that require investigation: 1) the elution order of the derivatives they identify is highly unlikely, and 2) the mass fragments they provide in their revised Table 3 is inconsistent with what would be expected from the methods provided in section 4.8.
- I do apologize for supplying the supplement information for the methylation analysis of this part we sent to analyze at Gangneung Wonju National University’s center lab so that we could take a long time to get the raw data and chromatogram that showed the mass fragments of each linkage. As we did the polysaccharide structural analysis for the previous publication as below, we did base on the standard analysis. As the reviewer mentioned in QA-1, 2
- I would show my previous chromatogram of the methods we used and how the interpretation how linkage we interpreted.
For example, we study monosaccharide and methylation analysis as the method described below.
- Surayot U., Lee JH., Kanongnuch C., Peerapornpisal Y., Park WJ., You SG. 2016. Structural characterization of sulfated arabinans extracted from Cladophora glomerata Kützing and their macrophage activation. Bioscience, Biotechnology, and Biochemistry. 80, 972–982.
The monosaccharide analysis
Gas chromatography-mass spectrometry (GC-MS) was used to examine the mon-osaccharide composition of C. glomerata Kützing polysaccharides. The sample was hydrolysed with trifluoroacetic acid (TFA, 4M) at 100 °C for 6 h, then reduced in water with sodium bo-rodeuteride (NaBD4) and acetylated with acetic anhydride. Finally, the sample was an-alysed by GC–MS (6890 N/MSD 5973, Agilent Technologies, Santa Clara, CA, USA) coupled with HP-5MS capillary column (30 m x 0.25 mm x 0.25 mm). As a carrier gas, nitrogen was applied. To certify the monosaccharide content, monosaccharide standards were employed.
- As we had a GC-MS chromatogram of monosaccharide, and we get the monosaccharide as a percent report.
- As we had a GC-MS chromatogram of linkage, and we get the linkage as a percent report.
In addition, in our previous studies associated with the immune regulation of polysaccharides that did not show a GC-MS chromatogram.
Many studies associated with the immune regulation of polysaccharides did not show a GC-MS chromatogram similar to our current study.
Comments on the Quality of English Language
Minor editing required
- It was corrected.
- Thanks.

Reviewer 4 Report
Comments and Suggestions for Authors
There are several points for revision:
1. Table 3 is not readable because of overlapped text.
2. Lines 300-302, 372-374: there is no evidence that all mentioned units are in the backbone. Moreover, terminal units (mannose) should be in the side chanis, not in the backbone. If some glucose units are sulfated at O-6 of some glucose units, the backbone could be of 1,3-linked glucose partially sulphated at O-6. However, it is still unclear about 1,6-linked galactose units. It should be a branched polysaccharide, and these and other units can be in the side chains.
3. Lines 382-392: Comparison with fucoidan from brown seaweeds is irrelevant because the active fraction contained only a low amount of fucose.
Comments on the Quality of English Language
The English grammar and style of the manuscript, especially new inserts, should be carefully corrected.
Author Response
Response to Reviewer 4 Comments
There are several points for revision:
- Table 3 is not readable because of overlapped text.
- We revised Table 3 as per the suggestion. Thank you very much.
- Lines 300-302, 372-374: there is no evidence that all mentioned units are in the backbone. Moreover, terminal units (mannose) should be in the side chanis, not in the backbone. If some glucose units are sulfated at O-6 of some glucose units, the backbone could be of 1,3-linked glucose partially sulphated at O-6. However, it is still unclear about 1,6-linked galactose units. It should be a branched polysaccharide, and these and other units can be in the side chains.
- Thank you for the precious comments.
- I do agree with the reviewer that the backbone could be of 1,3-linked glucose partially sulfated at O-6 because, after desulfation, (1→3)-linked glucose significantly increased concomitantly with a decrease in (1→3,6)-linked glucose. Moreover, the 1→6-linked galactose units could be a branched polysaccharide, and other units and terminals can be in the side chains. Therefore, future studies would be required on the glycosidic linkages of JF3, which were confirmed by 1D and 2D NMR analyses, to investigate not only the anomeric but also the position of the connecting on each linkage. Moreover, FT-IR analysis is required for the sulphate groups, carboxyl groups, and amide groups. The investigation into the molecular weight and fine structures is essential to gain a deeper understanding of the relationships between structural elements and anti-inflammatory activity to accomplish their maximum effects, which could be conducted in future studies.
- The objective and scope of this manuscript are to study on the fractionation of polysaccharides from smithii before investigating the protective action of L. smithii upon inflammation triggered by LPS and probable signal pathways implicated through in vitro studies using RAW264.7 cells. Furthermore, the polysaccharides were also evaluated for physiochemical and molecular characteristics. Please kindly consider this manuscript for publication in Marine Drugs based on the available data presented within the manuscript. Thank you for taking the time to review our work.
- Lines 382-392: Comparison with fucoidan from brown seaweeds is irrelevant because the active fraction contained only a low amount of fucose. However, the exact structure of this polysaccharide has not been established, and additional analyzes are needed to complete the structural analysis of the most active fraction. The issues for revision are summarized below.
- Regarding the suggestion made by other reviewers suggesting adding information about fucose, since it appears as an anti-inflammatory compound in several in vitro and in vivo experimental models. In our results, fucose is present in a minor proportion (0.10%); sometimes, the minor components are the most active alone or in combination with others. Therefore, we included information about fucose and anti-inflammatory activity in the discussion section.
- The English grammar of the manuscript was corrected by the native speaker, especially the new inserts.
- Thank you for your time to review this manuscript.
Yours sincerely,
Utoomporn Surayot, Ph.D.
Assistant Professor
College of Maritime Studies and Management
Chiang Mai University
Samut Sakhon 74000, Thailand
Round 3
Reviewer 2 Report
Comments and Suggestions for Authors
You must supply the supplementary data requested for the work to be reviewed.
Comments on the Quality of English Language
Minor edits.
Author Response
As reviewer 2 suggested that I supply the supplementary data, including the chromatogram and mass spectra, I did the methylation analysis of JF3 again, and the results (chromatogram and mass spectra) are in the attached file.
Yours sincerely,
Utoomporn Surayot, Ph.D.
Assistant Professor
College of Maritime Studies and Management
Chiang Mai University
Samut Sakhon 74000, Thailand

Round 4
Reviewer 2 Report
Comments and Suggestions for Authors
I reiterate my previous concerns about the linkage analysis results. In this latest version the authors have provided mass spectra of the PMAA derivatives. The mass spectrum shown for Araf-(1- is not correct; this is mass spectrum expected for a Hexp-(1-. The other spectra provided are correct for the PMAAs identified.
Further, the terminal/branch point ratio (0.08) shows there is a high degree of sulphation (JF3 has almost 23% sulphate). The linkage analysis should be repeated after desulphation to show the positions of the sulphate groups. Is the polysaccharide composed of 3-linked hexoses with 6-O-sulphation, or 6-linked hexoses with 3-O-sulphation?
The major concerns must be addressed.
Comments on the Quality of English Language
Adequate
Author Response
- Regarding the suggestion made by reviewer 2 in the previous revision to provide supplementary data, including chromatograms and mass spectra, we conducted methylation analysis of JF3 without desulfation, as previously reported. We have included the results, including chromatograms and mass spectra, as requested. Moreover, reviewer 2 has requested to repeat the linkage analysis after desulfation to show the positions of the sulfate groups. However, to the best of our knowledge, this manuscript already includes sufficient data on the primary structure of polysaccharides extracted from Lobonema smithii The positions of the sulphate groups, which are important information for the experiments, are ongoing now. We study the structure and activity relationships, which also study functional groups (sulphate) or molecular weight, which exactly affects the anti-inflammatory activity. Therefore, the objective and scope of this manuscript is a study on the fractionation of polysaccharides from L. smithii, before investigating the protective action of L. smithii upon inflammation triggered by LPS and probable signal pathways implicated through in vitro studies using RAW264.7 cells. Furthermore, the polysaccharides were also evaluated for physiochemical and molecular characteristics. Please kindly consider this manuscript for publication in Marine Drugs based on the available data presented within the manuscript. Thank you for taking the time to review our work.
- Thank you for your time to review this manuscript. Thank you very much.
Yours sincerely,
Utoomporn Surayot, Ph.D.
Assistant Professor
College of Maritime Studies and Management
Chiang Mai University
Samut Sakhon 74000, Thailand
